# The Danish Version of the 20-Item Prosopagnosia Index (PI20): Translation, Validation and a Link to Face Perception

**DOI:** 10.3390/brainsci13020337

**Published:** 2023-02-16

**Authors:** Erling Nørkær, Ester Guðbjörnsdóttir, Sofie Black Roest, Punit Shah, Christian Gerlach, Randi Starrfelt

**Affiliations:** 1Department of Psychology, University of Copenhagen, 1353 København, Denmark; 2Department of Psychology, University of Bath, Bath BA2 7AY, UK; 3Department of Psychology, University of Southern Denmark, 5230 Odense, Denmark

**Keywords:** developmental prosopagnosia, face recognition, face perception, object recognition, self-report measures

## Abstract

Developmental prosopagnosia (DP) is a neurodevelopmental condition characterized by face recognition problems. Psychometrically sound self-report measures of face recognition problems are important tools in classification of DP. A widely used measure of such problems is the 20-item prosopagnosia index (PI20). Here, we present a Danish translation of the PI20 (PI20_DK_). We administered the PI20_DK_ alongside three objective measures of face and object processing performance to 119 participants to validate the PI20_DK_. Further, we assess the underlying factor structure of the PI20_DK_. Finally, as the first study in the field, we investigate the association between self-reported face recognition ability and face perception performance. The project was preregistered prior to data collection. The results suggest excellent convergent validity, discriminant validity and internal consistency for the PI20_DK_. A confirmatory factor analysis, however, indicates a suboptimal fit of the PI20_DK_ to a one factor solution. An investigation of the association between the PI20_DK_ and face perception suggests that the poor fit may reflect that the PI20_DK_ measures problems with face recognition in general and not specifically face memory problems.

## 1. Introduction

Developmental prosopagnosia (DP) is a neurodevelopmental condition characterized by lifelong problems with face recognition. Individuals with DP struggle to recognize personally familiar faces such as those of friends and family, and they have problems with perception and discrimination of unfamiliar faces [1]. People with DP often rely on overt strategies for face recognition such as memorizing distinctive facial features (e.g., unusual eyebrows) or using extra-facial cues to identity (e.g., voice or gait) [2].

For a long time, DP was considered to be a rare condition, but over the last 10–15 years, it has become clear that it is in fact rather common [3]. The best estimates to date suggest that about 2% of the population have severe problems with face recognition [4,5], to a degree that affects their everyday life. This has led to an increased interest in diagnostic tools for prosopagnosia. Self-report measures of face recognition ability are considered an important tool in classification of DP [6]. Some studies rely on self-reports to screen potential DP individuals [7], while others combine self-report measures and objective assessments of face recognition such as the Cambridge Face Memory Test (CFMT) to classify individuals with DP (e.g., [8]). Indeed, Dalrymple & Palermo [9] suggest that a DP diagnosis should be based on both performance on tests of face memory as well as a “subjective feeling of repeated face recognition failures in daily life” (p. 75). This diagnostic emphasis on the subjective account in DP highlights the importance of a standardized, valid self-report measure of the degree of face recognition difficulties (see also [10]).

Given that different research groups have historically relied on different self-report measures of face recognition ability and criteria for classification of DP, it is difficult to compare results across studies. To meet the demand of a standardized self-report measure for DP classification, Shah and colleagues [11] developed the 20-item prosopagnosia index (PI20). The PI20 questionnaire has since been widely used in DP-studies alongside measures of objective face memory performance [12,13,14,15], and has been demonstrated to play a useful role in the identification of individuals with DP [16]. In calling for homogenous diagnostic criteria across the DP research field, Corrow et al. [17] suggested using the PI20 as a diagnostic inclusion criterion indicating difficulty with faces in everyday life. This study aims to validate the Danish version of the PI20 (PI20_DK_).

The PI20 was originally developed and validated in English and has largely been used in samples of English speakers. Portuguese and Chinese versions of the questionnaire have also been developed and validated [18,19]. The purpose of this study is to translate and validate a Danish adaptation of the PI20 [11], in order to be able to use the PI20 in samples of individuals with Danish as their first language. We translated the PI20 following a protocol for translation and adaptation of patient-reported outcome measures [20].

In order to validate the translated version (PI20_DK_), we aimed to establish indications of good validity and reliability in a Danish setting. Specifically, we were interested in assessing the *convergent validity, discriminant validity, internal consistency* and the assumption of *unidimensionality* of the PI20_DK_. To this end, we administered the PI20_DK_ alongside three objective measures of face and object processing in a sample of Danish university students. The three measures were the Cambridge Face Memory Test (CFMT), the Cambridge Car Memory Test (CCMT) and the Cambridge Face Perception Test (CFPT) [21,22,23], three tests commonly used in studies of DP. The CFMT and CCMT were used to establish construct validity of the PI20_DK_, while the role of the CFPT was to determine the degree to which the PI20_DK_ measures the perceptual aspect of face recognition ability. To our knowledge, the link between self-reported face recognition ability as measured by the PI20 and objective face perception performance has not been investigated in previous research^1^. Mishra et al. [24] investigated whether CFPT scores predict PI20 scores, but in a multiple regression model with both CFPT and another face perception measure as predictors of PI20 based on data from 30 DP individuals and 30 matched controls. As such, the present approach is a novel investigation of the link between self-reported face recognition ability and objective face perception performance in a non-clinical sample.

In short, the present study aims to:Validate the PI20_DK_ for use with Danish speaking individualsAssess the underlying factor structure of the PI20_DK_Investigate the degree to which the PI20_DK_ measures problems with face memory and face perception, respectively

In the following, our hypotheses and methods are introduced. All hypotheses and analyses were pre-registered prior to data collection. The results and discussion sections are structured to address each hypothesis separately.

## 2. Materials and Methods

### 2.1. Preregistration and Hypotheses

The study was preregistered on Open Science Framework (OSF) (see Preregistration) before data were collected. Here, we specified eight hypotheses to test the validity, reliability and underlying factor structure of the PI20_DK_ as well as its association with face perception performance.

**Hypothesis** **1.**
*Convergent validity of the PI20_DK_. Total scores on the PI20_DK_ will be negatively associated with total scores on the CFMT.*


**Hypothesis** **2.**
*Discriminant validity of the PI20_DK_. Total scores on the PI20DK will be unrelated to total scores on the CCMT.*


**Hypothesis** **3.**
*Internal consistency of the PI20_DK_. The internal consistency as measured by Cronbach’s alpha will reach at least acceptable levels (> 0.70) as prescribed by e.g., Cortina (1993) [25].*


**Hypothesis** **4.**
*Internal consistency of the CFMT and the CCMT. The CFMT and CCMT will both show at least acceptable Cronbach’s alpha levels (>0.70).*


**Hypothesis** **5.**
*Item-total correlations of the PI20_DK_. Each of the 20 items of PI20_DK_ will have a positive association with the total score.*


**Hypothesis** **6.**
*Difference between relations between PI20_DK_ and car memory and PI20_DK_ and face memory. The correlation coefficient for PI20_DK_-CFMT will be larger than for PI20_DK_-CCMT.*


**Hypothesis** **7.**
*The underlying factor structure of PI20_DK_. A confirmatory factor analysis (CFA) with one extracted factor will confirm that the underlying variable structure of the PI20_DK_ fits a one factor solution (the assumption of unidimensionality).*


**Hypothesis** **8.**
*Relation between PI20_DK_ and face perception. Higher scores on PI20_DK_ will be associated with lower accuracy scores on CFPT.*


### 2.2. Participants

An a priori power analysis (explained in detail in the preregistration of the study) indicated that 92 participants would ensure at least 90% power for the analysis of convergent validity, assuming the correlation coefficient between the PI20_DK_ and the CFMT would be similar to those found in prior research [19,26]. We aimed to include at least 120 participants to allow plenty of room for dropouts and otherwise unusable data. 

To test the eight hypotheses, we collected PI20_DK_ and behavioral data from a cohort of Danish University students who participated as part of their course in cognitive psychology. 139 individuals participated in the study. A total of 20 participants did not complete all four measures in the standard setup, and their data were excluded, so all analyses were performed only on participants with available scores on all four tasks. Of the final sample (*N* = 119) 84 identified as female, 35 as male; 107 were right-handed, 5 were left-handed and 7 reported to be ambidextrous. Additional demographic data, such as age and ethnicity, were not collected due to local data protection regulations. However, since our participants were all University students, it is a convenience sample and not intended to be representative of the general Danish population. While this lack of sample diversity and representativity may constrain the generalizability of the results, the sample was suitable for the current study. The primary purpose of this study is to make a case for the PI20_DK_ as a psychometrically sound measure, rather than making claims about face recognition ability in the general population. In fact, the homogeneity of the present sample may be an advantage for the current purpose, as the samples of the Portuguese and Chinese validation studies were similarly recruited from populations of University students [18,19].

The participating students were naïve to the hypotheses. The course in which the testing took place is approved by the study board at the Department of Psychology, University of Southern Denmark, and the experiments conducted do not require additional ethical approval according to Danish Law and the institutional requirements. The students were informed prior to participation that the collected data might be used in an anonymous form in research publications. Participants were free to terminate their participation at any time.

### 2.3. Procedure

Data were collected over four days with each participant completing all measures in the same day over the course of approximately 1 h and 45 min. 

Participants completed the following measures in the following order:(1)PI20_DK_–participants filled out an online version of the Danish adaptation of the original PI20 [11];(2)Cambridge Face Memory Test (CFMT)–upright faces [27];(3)Cambridge Car Memory Test (CCMT)–upright cars [21];(4)CFMT with inverted faces;(5)CCMT with inverted cars;(6)Cambridge Face Perception Test (CFPT) [22].

The task order of CFMT and CCMT was counterbalanced so that half of the participants completed CFMT before CCMT and vice versa. All participants began with the two upright stimuli test forms ((2) and (3)). Data from the inverted versions of CFMT and CCMT ((4) and (5)) were not part of the present study.

Participants were given a 20 min break before the CFPT. 

### 2.4. Measures

**PI20_DK_.** The original PI20 is a twenty-item questionnaire developed to assess difficulties with face recognition and aid in the classification of DP cases [11]. The 20 items are statements regarding different aspects of trouble with face memory (e.g., ‘I often mistake people I have met before for strangers’ and ‘I have to try harder than other people to memorize faces’). Participants are asked to rate each statement on a Likert scale ranging from 1 (Strongly disagree) to 5 (Strongly agree).

We translated the PI20 into Danish following guidelines of good practice for translation and cultural adaptation [20]. The guidelines were followed in order to ensure consistency and comparability between the PI20_DK_ and the original PI20. Prior to the translation from English to Danish, permissions for copyright material were checked and the original author, Punit Shah, was invited to take part in the process.

Initially, the original 20 English items were translated to Danish. In order to avoid bias related to personal writing styles and potential misinterpretations, two independent native Danish speakers conducted the forward translations. The forward translations were compared and merged into a single forward translation. This process focused on reconciling discrepancies between the two initial translations regarding individual speech habits and preferences, ambiguities and making the translation easily understandable to a Danish audience. Two native English speakers then back translated the reconciled forward translation independently. During the back translation, a focus was kept on conceptual equivalence rather than literal translation. Next, the two translations were compared with the original English version, where semantic, idiomatic, experiential, and conceptual equivalence were taken into consideration.

After the translation, an online cognitive debriefing survey was made to check the understandability, interpretation, and cultural relevance of each item. In the debriefing, six native Danish speakers filled out the PI20_DK_. For each item, the respondents were asked whether the meaning was clear and to describe each item in their own words. On the basis of the debriefing results, minor adjustments were made. This resulted in a final Danish version of PI20; the PI20_DK_. The final version is available as Appendix A on OSF alongside a document detailing the forward translation, back translation and revision process (see Appendix A).

Participants completed the PI20_DK_ physically present at the test site but via an online platform. Raw scores on 5 items were reversed (Items 8, 9, 13, 17 and 19) so that higher scores indicated more problems with face recognition on all 20 items. Raw scores were summed to form total PI20_DK_ scores for each participant.

**CFMT.** The Cambridge Face Memory Test is a widely used test of face recognition ability [23]. The CFMT was used to validate the original PI20 [11] and further research has shown a strong negative association between PI20 scores and number of correct responses on the CFMT (*r* = −0.39 in Gray et al. [26] and *r* = −0.43 in Ventura et al. [19]). A negative association in this context implies that higher scores on the PI20 (indicating more problems with face recognition) is associated with worse CFMT performance (fewer correct responses). We therefore included the CFMT to assess the convergent validity of the PI20_DK_ (Hypothesis 1). In the CFMT, participants are consecutively shown three images of six different faces from different angles. Over the course of 72 trials distributed in three conditions of increasing difficulty (practice/no noise/noise), participants are asked to identify the learned target face among distractor faces. The total number of correctly answered trials for each participant was used in the statistical analyses. 

**CCMT.** The Cambridge Car Memory Test is equivalent to the CFMT but with cars instead of faces as stimulus material [21]. In the original PI20 validation study no association between CCMT performance and PI20 scores was observed (Pearson’s *r* = −0.068), indicating that the PI20 specifically taps problems with face memory, not just object memory in general. Therefore, the CCMT was included here to assess the discriminant validity of the PI20_DK_. Similar to CFMT, the total number of correctly answered CCMT trials for each participant was used in the statistical analyses.

**CFPT.** The Cambridge Face Perception Test is a sorting task designed to measure perception of facial similarity [22]. Across 16 trials, participants are asked to sort six faces based on their resemblance to a target face. In half of the trials, the faces are upright, in the other half inverted. Only performance on upright trials were used in this study. For each trial, the distance between the sequences of faces produced by the participants and the correct sequence is recorded as a measure of impaired face perception. This total deviation from the correct position of each upright face was transformed to a measure of accuracy by the following formula: $100*\left(1-\left(\frac{UprightDeviation}{144}\right)\right)$, and this accuracy measure was used in all statistical analyses involving the CFPT. To our knowledge, the correlation between CFPT performance and PI20 scores has not been investigated in previous research. As such, the present study is the first in the field to assess the link between self-reported face recognition ability and objective face perception performance.

### 2.5. Statistical Analyses

Data were prepared and analyzed in R version 4.2.1 [28]. Raw data and the script used for analyses are available on OSF (see Appendix A).

**Confirmatory analyses.** To investigate the convergent and discriminant validity of the PI20_DK_ (Hypotheses 1 and 2) we calculated Pearson correlation coefficients between PI20_DK_ total scores and CFMT and CCMT total scores, respectively. We assessed the internal consistency of the PI20_DK_, the CFMT and the CCMT (Hypotheses 3 and 4) by calculating Cronbach’s alpha values for the three measures. The degree to which all items of the PI20_DK_ measure the same construct was assessed by calculating item-total correlations (Pearson’s *r*) for all 20 items (Hypothesis 5). To determine if there was a significant difference between the correlations between the PI20_DK_ and the CFMT and the CCMT, respectively (Hypothesis 6), we compared the two *r* values using the *r*-to-*z* transformation for dependent samples based on Hittner, May & Silver’s [29] modification of Dunn & Clark’s [30] *z* transformation. This transformation and comparison was performed using the cocor R package [31]. The underlying factor structure of the PI20_DK_ (Hypothesis 7) was assessed by fitting the data to a one factor solution with a CFA based on maximum likelihood estimation using the lavaan R package [32]. In line with recommendations by Kline (2015) [33] we extracted and evaluated the following fit indices for the CFA: Comparative Fit Index (CFI), Tucker-Lewis Index (TLI; [34]), Root Mean Square Error of Approximation (RMSEA) and Standardized Root Mean Square Residual (SRMR). Additionally, a Chi-square test was conducted on the CFA model to assess the discrepancy between the sample and fitted covariance matrices. Finally, to assess the relationship between face perception and the PI20_DK_ (Hypothesis 8) we calculated the Pearson correlation coefficient between CFPT accuracy scores and PI20_DK_ total scores.

**Exploratory analyses.** As an exploratory analysis, we included a comparison of the PI20_DK_-CFPT and PI20_DK_-CFMT correlation coefficients using the same *r*-to-*z* transformation as described above for the evaluation of hypothesis 6. Additionally, to qualify the analyses involving the CFPT, we included an exploratory Cronbach’s alpha analysis for the measure. 

## 3. Results

Means, standard deviations and ranges of the study’s four measures (PI20_DK_, CFMT, CCMT and CFPT) and all PI20_DK_ items are presented in Table 1. 

The associations between PI20_DK_ total scores and scores on the three performance tests are presented in scatter plots in Figure 1. PI20_DK_ total scores correlated significantly with performance on the CFMT (*p* < 0.001) and the CFPT (*p* = 0.023), but not with CCMT performance (*p* = 0.907). (Note that there were two outliers on the CFPT. These data were not excluded as per the preregistration. However, if these data points are excluded from the analysis, the correlation with PI20_DK_ is *r* = −0.17, *p* = 0.072). The *r*-to-*z* transformed difference between the PI20_DK_-CFMT and PI20_DK_-CCMT correlation coefficients was significant (*z* = 2.84, *p* = 0.005, two-tailed). The exploratory analysis of the *r*-to-*z* transformed difference between the PI20_DK_-CFMT and PI20_DK_-CFPT correlation coefficients was not significant (*z* = 1.30, *p* = 0.19, two-tailed).

Cronbach’s alpha values for PI20_DK_, CFMT, CCMT and CFPT (upright trials) were all in the 0.81–0.90 range (see Table 2). The 20 item-total correlations were all significantly larger than zero (Table 3).

The CFA Chi-square test was significant (*p* < 0.001) and three of the four reported fit indices were outside the cutoff values (Table 4).

## 4. Discussion

The aim of the present study was to translate the PI20 questionnaire to Danish and validate this measure in a Danish sample. The PI20 is widely used in studies of Developmental Prosopagnosia (DP). Although the questionnaire is based on the subjective report of face recognition abilities, the original version has been shown to be a valid and useful tool in the detection and classification of DP [11,16,17]. In order to validate the PI20_DK_ we initially formulated eight pre-registered hypotheses regarding the relationship between the PI20_DK_ and two other measures of face processing: a memory (CFMT) and a perception test (CFPT), and an object memory test (CCMT). The results supported seven of the eight preregistered hypotheses. In the following, we discuss each hypothesis to establish the PI20_DK_ as a valid and reliable measure of problems with face recognition in a Danish context. Additionally, we discuss the underlying factor structure of the PI20_DK_ and the relationship between the PI20_DK_ and face perception.

### 4.1. Validity of the PI20_DK_

If the PI20_DK_ is a valid measure of problems with face recognition, it is reasonable to expect its scores to correlate with performance on an objective measure of face memory such as the CFMT (Hypothesis 1). A negative correlation would indicate that higher scores on the PI20_DK_ (more self-reported problems with face recognition) is associated with lower performance on the CFMT (fewer correct responses). We expected the size of Pearson’s *r* for the relation between the CFMT and the PI20_DK_ to be on level with other studies examining this link (*r* = −0.39 in Gray et al. (2017) and *r* = −0.43 in Ventura et al. (2018) [19,26]). The correlation in the present sample was slightly weaker (*r* = −0.34) but highly significant (*p* < 0.001) and may be interpreted as a sign of good convergent validity (Table 2).

As an indication of discriminant validity, we expected the Pearson correlation between the CCMT and the PI20_DK_ to be non-significantly different from 0 (Hypothesis 2). This was based on the assumption that for the PI20_DK_ to be a valid measure of problems with faces specifically, its scores should not be associated with performance on memory of non-face objects. In the validation study of the original PI20 no association with CCMT performance was observed (*r* = −0.07; [11]). Similarly, in the present sample, the PI20_DK_-CCMT correlation was practically nonexistent (*r* = 0.01, *p* = 0.907) indicating good discriminant validity (Table 2). It should be noted that the support of this hypothesis is not a claim that impaired non-face object recognition is not involved in DP at all. Several studies [35,36] and a comprehensive review [37] have found impaired non-face object recognition in samples of DP participants. The claim here is entirely psychometric in the sense that for the PI20_DK_ to be a valid, unidimensional measure of problems with face recognition, its scores should not be associated with performance on non-face object processing.

To strengthen the notion that the PI20_DK_ measures problems with face recognition and not just general object recognition, we expected a significant difference between the PI20_DK_-CFMT and the PI20_DK_-CCMT Pearson correlation coefficients (*r* = −0.34 and *r* = 0.01, respectively) (Hypothesis 6). This difference was indeed significant (*z* = 2.84, *p* = 0.005, two-tailed), further indicating strong validity for the PI20_DK_ as investigated with the first two hypotheses.

In order to justify the use of correlations with CFMT and CCMT as measures of convergent and discriminant validity, respectively, we expected the two measures to show at least acceptable Cronbach’s alpha levels (0.70) (Hypothesis 4). The two tests have shown good internal consistency in past research with alpha values in the 0.89–0.90 range for the CFMT [5,38,39] and 0.84 for the CCMT [21]. In the present sample, the alpha values were 0.88 and 0.81 for CFMT and CCMT, respectively, indicating acceptable levels of internal consistency (Table 2). As such, the indications of convergent and discriminant validity discussed above are not endangered by poor reliability estimates of the CFMT and CCMT.

### 4.2. Reliability of the PI20_DK_

As an indication of good reliability, the PI20_DK_ should exhibit at least an acceptable (>0.70) Cronbach’s alpha value (Hypothesis 3). We expected the alpha value of PI20_DK_ to reach levels similar to the ones reported in the original validation study of the PI20 (α = 0.96, *N* = 319; [11]) and for the Chinese adaptation (α = 0.929, *N* = 647; [18]). In the present sample the alpha value was more modest (α = 0.90) but still well above an acceptable level, indicating good internal consistency of the PI20_DK_ scale (Table 2).

We expected all item scores to show significant correlations with the total PI20_DK_ score as an indication that they all measure problems with face recognition (Hypothesis 5). The Chinese PI20 validation study [18] found significant, positive item-total correlations for all items except item 3 (*r* = −0.036). In the present sample, scores on all 20 items had positive, significant associations with the total score (Table 3). However, this association was by far weakest for item 3 (*r* = 0.19), which—with the results from Sun et al. (2021) in mind—may suggest that item 3 is not part of the same latent variable as the 19 other items.

### 4.3. PI20_DK_ and Face Perception

To our knowledge, the correlation between PI20 and CFPT has not been reported in other validation studies. One might argue, that if the PI20_DK_ exclusively measures self-reported face *memory* ability, the PI20_DK_-CFPT correlation should be non-significantly different from zero, indicating discriminant validity, since the CFPT operates with face stimuli that are unknown to the participant and does not require memorization and recognition. On the other hand, face perception is surely an essential component in recognizing faces, and if the PI20_DK_ measures self-reported face recognition ability in general, one would expect higher PI20_DK_ scores to be associated with lower CFPT accuracy scores. Due to this ambiguity, we did not include the PI20_DK_-CFPT correlation as a measure of either discriminant or convergent validity. The aim of including the association between the PI20_DK_ and the CFPT in the analyses was simply to explore to which degree the PI20_DK_ taps perceptual aspects of face recognition (Hypothesis 8).

The negative, significant correlation between the PI20_DK_ and the CFPT upright scores in the present sample (*r* = −0.21, *p* = 0.023, two-tailed; Table 2) may indicate that the PI20_DK_ measures not only problems with face memory, but also face perception. The lack of a difference between the PI20_DK_-CFMT and PI20_DK_-CFPT correlation coefficients discovered by exploratory analysis further indicates that high scores on the PI20_DK_ might reflect both perception and face memory problems. It should be noted that the comparison of the two correlation coefficients is corrected by the internal correlation between CFMT and CFPT, so the exploratory finding cannot be explained simply by the fact that face memory and face perception as measured by CFMT and CFPT are intertwined constructs. It should be noted that two outliers were included in this analysis, as per the preregistration protocol, and that the results shift somewhat if these participants are excluded. This underlines the exploratory nature of this finding, which clearly warrants replication in independent studies.

The observation that the PI20 _DK_ correlates with both face perception and memory performance does not undermine the PI20 measure as a diagnostic tool for DP, since both perception and memory deficits are commonly reported in DP [40]. The question is rather if it poses a psychometric problem for the PI20_DK_, as it might challenge the assumption of unidimensionality of the scale. The summation of the 20 item scores to a total score rests on the assumption that all items measure the same thing. If the PI20_DK_ (and, perhaps, the PI20) measures more than one construct, it should consist of more than one scale. However, it may be the case that the latent variable of the PI20_DK_ (and the PI20) is face recognition ability in general, and that both face perception and face memory are essential components in face recognition. What seems to be two distinguishable constructs—face perception and face memory—may be equally important nodes in a face recognition network. As such, the assumption of unidimensionality may hold for the PI20_DK_ as a measure of problems with face recognition in general, but not as a measure of problems with face *memory* specifically. In the following section, we further discuss the assumption of unidimensionality of the PI20_DK_ and point to possible solutions to the problem constituted by the results.

### 4.4. The Underlying Factor Structure of PI20_DK_

The original PI20 validation study found indications of a strong single factor structure accounting for 61% of the variance by means of exploratory factor analysis [11]. On that basis, we expected a CFA to show a good fit with a one factor solution, supporting the assumption of unidimensionality of the full PI20_DK_ scale (Hypothesis 7). Sun et al. (2021) found such support in their validation of the Chinese PI20 with all reported fit indices showing a good fit with a one factor solution [18]. In the present sample, however, the CFA fit indices did not support a one factor solution. The Chi-square test was highly significant (*p* < 0.001) and three of the four fit indices (RMSEA, CFI and TLI) were outside the acceptable ranges as suggested by Hu & Bentler [41] (Table 4). RMSEA is an absolute fit index that assesses how far the proposed one factor solution is from a perfect model, while CFI and TLI are incremental fit indices that compare the proposed one factor model to a baseline model [42]. Hu & Bentler [41] suggested that the RMSEA should be lower than 0.06, while CFI and TLI should each exceed 0.95 to indicate an acceptable fit for a CFA based on maximum likelihood estimation of continuous data. The values in the present sample are outside these acceptable ranges (RMSEA = 0.09, CFI = 0.84, TLI = 0.82) which indicates that data are incompatible with the proposed one factor solution, bringing into question the assumption of unidimensionality of the PI20_DK_.

There are several possible explanations for why the PI20_DK_ data does not seem to fit a one factor solution. First, a general criticism of the cutoffs for the fit indices is appropriate. Our evaluation of fit indices is rather conservative, as it is based on the cutoff values from Hu & Bentler [41]. It has been argued that CFA goodness-of-fit should be evaluated based on dynamic fit index cutoffs rather than static values as performed here [43]. Further, the lack of a good fit may be attributed to the modest sample size (*N* = 119). Generally, larger samples would be preferable for CFA purposes, but we deemed it appropriate on the basis that the number of items is relatively small and the items-to-factor ratio is large [44]. The Chinese validation study had a substantially larger sample (*N* = 647; [18]), which may have made it more likely to detect a good fit with a one factor solution. The participants in the Chinese validation study were predominantly young females as was the case in the present sample. However, the cultural differences between a Danish and Chinese population may pose an issue for comparing results and expecting similar patterns across countries.

A third and perhaps the most parsimonious explanation for the less-than-optimal CFA fit is that the assumption of unidimensionality of the scale does not hold, i.e., that the PI20_DK_ measures more than one construct. Indeed, looking at the item content, it is not unreasonable to intuitively question the notion of unidimensionality. For instance, Item 3 (‘I find it notably easier to recognize people who have distinctive facial features’) stands out as an item that even people without face recognition problems would agree with. Consulting the descriptive statistics for Item 3 (Table 1) supports this notion, as Item 3 has the highest mean score (3.76), indicating that it is very common to find it easier to recognize people who have distinctive facial features. Looking at the item-total correlations (Table 3), Item 3 has the markedly lowest association with the total PI20_DK_ score (*r* = 0.19), further indicating that it might not measure the same latent variable as the other items–at least not to the same extent. The only item with similarly low item-total correlation in the present sample was Item 13 (‘I am very confident in my ability to recognize myself in photographs’) (*r* = 0.24), which may reflect that self-face recognition is comparatively less impaired in DP [45], so people may generally agree with this item regardless of problems with face recognition. This indeed seems to be the case in the present sample, in which the mean value of Item 13 was 1.43 (Table 1). Interestingly, the item-total correlations for Item 3 was not significantly different from 0 in the Chinese validation study of PI20 (*r* = −0.04), and the item-total correlation for Item 13 (*r* = 0.43) was markedly lower than the remaining 18 values (all *r*’s > 0.53) [18]. As such, Item 3 and Item 13 may not have ‘problems with face recognition’ as their latent variable, challenging the assumption of unidimensionality.

Questioning the assumption of unidimensionality warrants a discussion about what exactly the PI20_DK_ (and the PI20 more generally) purports to measure in the first place. The original PI20 was developed to assess the presence of prosopagnosic traits and aid in the diagnosis of DP [11]. However, several studies have indicated that DP is a heterogenous condition [46,47] and there is still a lack of consensus on how to define and classify DP [48]. As such, it may not even be a meaningful goal to show unidimensionality of the scale. Rather, it may be the case that the PI20 taps into several different latent variables, each of importance for the diagnosis of DP. The fact that the PI20_DK_ correlates with both face memory and face perception performance (assuming CFMT and CFPT measures memory and perception, respectively) may be an indication of this. Further investigations of the PI20 may be fruitful if they aim to either exclude items to bolster the notion of a unidimensional scale, or if they aim to map the different DP relevant components underlying the PI20 and PI20_DK_. This may result in either a shortening of the questionnaire or in a division of the scale into subscales that fit the latent variable structure better. These approaches may strengthen the diagnostic quality of the PI20 as a valid and reliable measure of the various subjective aspects of DP. This may ultimately improve our understanding of prosopagnosia and face recognition ability more generally.

## 5. Conclusions

The purpose of this study was to validate the Danish translation of the PI20. The data supported seven of the eight pre-registered confirmatory hypotheses, indicating excellent validity and reliability for the PI20_DK_. The assumption of unidimensionality of the scale was, however, challenged by the factor analysis results. This, in addition to the substantial association between PI20_DK_ scores and face perception performance, may suggest that the PI20_DK_ measures several important components of face recognition, not only face memory problems.

Future research should assess the sensitivity and specificity of the PI20_DK_ by determining how well it discriminates between people with and without face recognition problems in a normative sample of Danish speaking individuals, as it has been shown to do in a UK-sample [16]. This would also allow setting a DP cutoff score for the PI20_DK_ for using it diagnostically in conjunction with objective measures of face processing. Furthermore, a shortening of both the Danish and English questionnaire and/or a separation of the scale into subscales may be warranted based on item analyses and factor analyses in larger samples. Finally, further investigations of the link between self-reported face recognition ability and objective face perception performance may improve our general understanding of developmental prosopagnosia.

## Figures and Tables

**Figure 1 brainsci-13-00337-f001:**
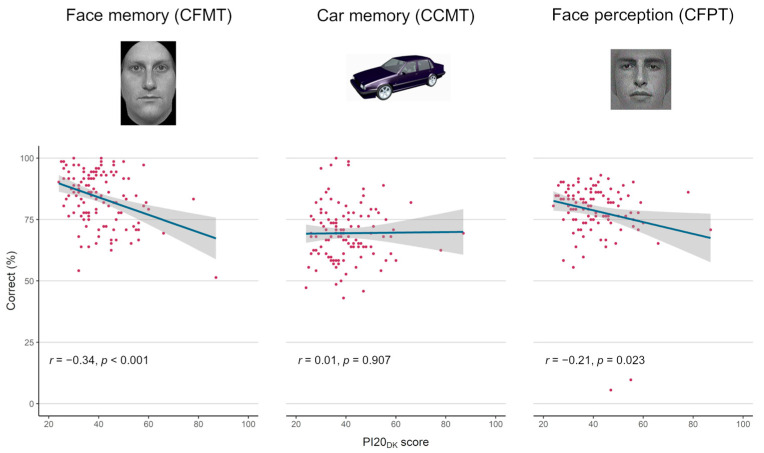
Scatter plots of scores on the PI20_DK_ and performance on the three objective measures (*N* = 119). The blue lines are linear regression models with 95% confidence intervals in grey. PI20_DK_ = the Danish version of the 20-item prosopagnosia index; CFMT = Cambridge Face Memory Test; CCMT = Cambridge Car Memory Test; CFPT = Cambridge Face Perception Test. The two extreme outliers on CFPT have been kept in the analysis, as specified in the preregistration. If these two points are excluded from the analysis, the result is *r* = −0.167, *p* = 0.072.

**Table 1 brainsci-13-00337-t001:** Descriptive statistics.

Variable	Mean	*SD*	Min	Max	*N*
CFMT total score	60.46	7.80	37	72	119
CCMT total score	49.98	7.99	31	72	119
CFPT accuracy	78.71	12.09	5.56	93.06	119
PI20_DK_ total score	40.18	10.49	24	87	119
PI20_DK_ item scores					
Item 1	2.16	0.90	1	5	119
Item 2	1.92	0.80	1	5	119
Item 3	3.76	1.05	1	5	119
Item 4	1.93	0.97	1	5	119
Item 5	1.24	0.64	1	5	119
Item 6	1.81	0.87	1	4	119
Item 7	1.55	0.95	1	5	119
Item 8 *	2.47	1.09	1	5	119
Item 9 *	3.31	1.09	1	5	119
Item 10	1.78	0.77	1	4	119
Item 11	1.33	0.68	1	4	119
Item 12	1.86	0.86	1	5	119
Item 13 *	1.43	0.80	1	5	119
Item 14	1.76	0.91	1	5	119
Item 15	1.40	0.76	1	5	119
Item 16	1.79	1.03	1	5	119
Item 17 *	2.54	1.15	1	5	119
Item 18	1.24	0.60	1	5	119
Item 19 *	3.05	0.98	1	5	119
Item 20	1.85	0.95	1	5	119

Note: PI20_DK_ = the Danish version of the 20-item prosopagnosia index; CFMT = Cambridge Face Memory Test; CCMT = Cambridge Car Memory Test; CFPT = Cambridge Face Perception Test. * Raw scores on these items were inverted so that higher scores indicated more problems with face recognition on all 20 items.

**Table 2 brainsci-13-00337-t002:** Internal consistency of PI20_DK_, CFMT and CCMT (N = 119).

Measure	Cronbach’s α
PI20_DK_	0.90
CFMT	0.88
CCMT	0.81
CFPT Upright	0.83 *

Note: PI20_DK_ = the Danish version of the 20-item prosopagnosia index; CFMT = Cambridge Face Memory Test; CCMT = Cambridge Car Memory Test; CFPT = Cambridge Face Perception Test. * The CFPT Cronbach’s alpha analysis was included as an exploratory analysis, i.e., it was not part of the preregistration.

**Table 3 brainsci-13-00337-t003:** Item-total correlations for the PI20_DK_ (N = 119).

Item Number	Pearson’s *r*	*p*-Value (Two-Tailed)
1	0.68	<0.001
2	0.76	<0.001
3	0.19	0.038
4	0.79	<0.001
5	0.49	<0.001
6	0.62	<0.001
7	0.73	<0.001
8	0.65	<0.001
9	0.54	<0.001
10	0.43	<0.001
11	0.46	<0.001
12	0.80	<0.001
13	0.24	0.009
14	0.61	<0.001
15	0.73	<0.001
16	0.74	<0.001
17	0.51	<0.001
18	0.57	<0.001
19	0.47	<0.001
20	0.71	<0.001

Note: PI20_DK_ = the Danish version of the 20-item prosopagnosia index.

**Table 4 brainsci-13-00337-t004:** Confirmatory Factor Analysis model fit indices (N = 119).

Fit Index	CFI	TLI	RMSEA	SRMR	*χ* ^2^
Value	0.84	0.82	0.09	0.08	324.35 *
Cutoff	>0.95	>0.95	<0.06	<0.08	*p* > 0.05

Note: CFI = Comparative Fit Index; TLI = Tucker-Lewis Index; RMSEA = Root Mean Square Error of Approximation. Cutoffs are based on Hu & Bentler [7]. * *p* < 0.001.

## Data Availability

Raw data (stripped of demographic variables due to data protection concerns), the Danish version of the 20-item prosopagnosia index, a document detailing the translation process as well as the R script used to analyze the present data are all available on Open Science Framework (OSF) and can be accessed via the following link: https://osf.io/rdfmw/files/osfstorage (accessed on 12 February 2023).

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
