# Peer review of "The Danish Version of the 20-Item Prosopagnosia Index (PI20): Translation, Validation and a Link to Face Perception"

_brainsci, 2023, doi:10.3390/brainsci13020337_

Round 1
Reviewer 1 Report
Authors were interested in assessing the convergent validity, discriminant validity, internal consistency and the assumption of uni-dimensionality of the PI20DK. To address this, authors administered the PI20DK alongside three objective measures of face and object processing in a sample of Danish university students. The three measures were the Cambridge Face Memory Test (CFMT), the Cambridge Car Memory Test (CCMT) and the Cambridge Face Perception Test (CFPT), three tests commonly used in studies of DP. The CFMT and CCMT were used to establish construct validity of the PI20DK, while the role of the CFPT was to determine the degree to which the PI20DK measures the perceptual aspect of face recognition ability.
Overall, the manuscript is well written, and the data is well presented but needs some improvements.
1)The introduction is poorly written. Make it more elaborative which will give comprehensibility.
2)Statistics and quantification are poorly presented. Authors should present data graphically. This will have more readability.
Author Response
Reviewer 1:
- Overall, the manuscript is well written, and the data is well presented but needs some improvements.
Response: Thank you, we have aimed to improve the paper based on your suggestions and those from Reviewer 2.
- The introduction is poorly written. Make it more elaborative which will give comprehensibility.
Response: We have rewritten the introduction, and made it more elaborate to ensure reader comprehension. In addition, we have changed the introduction as suggested by Reviewer 2
- Statistics and quantification are poorly presented. Authors should present data graphically. This will have more readability.
Response: We now include a figure with scatter plots showing the key analysis of the relationship between the PI20DK and the three behavioural measures. This illustrates both the relationship and the individual data points. We have also included a graphical abstract representing these key findings.
Reviewer 2 Report
This paper presents an academic study on developmental prosopagnosia (DP), the main aim of which is to validate the Danish translation of the PI20.
Overall:
- The study and the topic are very interesting.
- The work is well written and organized.
- The experiments are well done and several metrics have been considered.
However, I recommend that
- Authors should highlight the main contributions of this paper in the last part of the introduction section in the form of bullet points.
- Authors should present the organization of the paper in the last part of the introduction.
- It is recommended that authors consider another population (rather than Danish) in a future work.
- Authors should revise the manuscript carefully as it contains some grammatical errors.
- The authors should make another comparison with recently published studies.
Author Response
Response to reviewers:
Reviewer 2:
Comments and Suggestions for Authors
This paper presents an academic study on developmental prosopagnosia (DP), the main aim of which is to validate the Danish translation of the PI20.
Overall:
- The study and the topic are very interesting.
- The work is well written and organized.
- The experiments are well done and several metrics have been considered.
Response: Thank you for your useful evaluation of our manuscript.
However, I recommend that
- Authors should highlight the main contributions of this paper in the last part of the introduction section in the form of bullet points.
- Authors should present the organization of the paper in the last part of the introduction.
Response: These two points are now included at the end of the introduction.
- It is recommended that authors consider another population (rather than Danish) in a future work.
Response: We do not understand this comment. The paper presents a validation of a Danish translation of the PI20, and as such, testing a Danish population was imperative to validate the measure.
- Authors should revise the manuscript carefully as it contains some grammatical errors.
Response: We have now carefully checked the spelling and grammar in the manuscript and made changes accordingly. We hope we have caught all the errors noted by the reviewer.
- The authors should make another comparison with recently published studies.
Response: We have searched the literature again, and have found no new adaptation studies of the PI20 that are not already mentioned in the article. There is, however, the study of Tsantani et al. (2021), which discusses the diagnostic quality of the PI20, and demonstrates that the PI20 can play a useful role in the identification of individuals with developmental prosopagnosia. We now mention this study in the article, both in the introduction, discussion, and conclusion.
Ref: Tsantani, M., Vestner, T., & Cook, R. (2021). The Twenty Item Prosopagnosia Index (PI20) provides meaningful evidence of face recognition impairment. Royal Society open science, 8(11), 202062.
Round 2
Reviewer 2 Report
The manuscript can be accepted in the current form.